# Etiology, Clinical Presentation and Incidence of Infectious Meningitis and Encephalitis in Polish Children

**DOI:** 10.3390/jcm9082324

**Published:** 2020-07-22

**Authors:** Kacper Toczylowski, Ewa Bojkiewicz, Marta Barszcz, Marta Wozinska-Klepadlo, Paulina Potocka, Artur Sulik

**Affiliations:** Department of Pediatric Infectious Diseases, Medical University of Bialystok, Waszyngtona 17, 15-274 Bialystok, Poland; ewa.bojkiewicz@umb.edu.pl (E.B.); martbarszcz@gmail.com (M.B.); marta.wozinska-klepadlo@umb.edu.pl (M.W.-K.); paulina.potocka@umb.edu.pl (P.P.); artur.sulik@umb.edu.pl (A.S.)

**Keywords:** meningitis, encephalitis, incidence, enterovirus, tick-borne encephalitis, Lyme neuroborreliosis

## Abstract

Little is known about the causes and the frequency of meningitis and encephalitis in Poland. We did a retrospective single-center cohort study of children under 18 years old hospitalized with infectious meningitis or encephalitis. Incidence rates were calculated using collected data from patients from the North-East Poland only. A total of 374 children hospitalized between 1 January 2015 and 31 December 2019 were included in the study. A total of 332 (89%) children had meningitis, and 42 (11%) had encephalitis. The etiology of the infection was established in 331 (89%) cases. Enteroviruses accounted for 224 (60%) of all patients. A total of 68 (18%) cases were tick-borne infections. Bacterial pathogens were detected in 26 (7%) children. The median length of hospital stay for children with enteroviral meningitis was 7 days (IQR 7–9), increasing to 11 days (8–13) in those treated with antibiotics. The incidence of meningitis was estimated to be 32.22 (95% CI, 25.33–40.98) per 100,000 and that of encephalitis to be 4.08 (95% CI, 2.07–8.02) per 100,000. By the broad use of molecular diagnostic methods, we managed to identify etiology of the infection in the majority of children. Our data suggest that thorough diagnostics of central nervous system infections are needed to rationalize treatment.

## 1. Introduction

Since the introduction of conjugate vaccines against bacterial pathogens, the incidence of bacterial meningitis has been declining. This decline, in combination with the use of molecular diagnostics, has caused increased importance of viruses as a cause of meningitis and encephalitis [1].

Identification of the cause of the infection is important to improve clinical care, and to reduce the unnecessary use of antibiotics and antivirals. Patients with meningitis and encephalitis are often treated with antibiotics, what can extend their hospital stay [2]. Although acyclovir is beneficial in herpes simplex virus (HSV) or varicella zoster virus (VZV) encephalitis, its role in meningitis caused by these viruses is uncertain [3]. Additionally, it has no activity against enteroviruses. The use of molecular methods for diagnosis of the central nervous system (CNS) infections may allow for more targeted use of antiviral drugs and contribute to improvement in the global antibiotic stewardship [4,5].

Enteroviruses and herpesviruses are commonly detected in adults and in children, but considerable geographical variation exists in the cause of infections of the CNS across the world [6,7]. Recent trends in infections of the CNS in Poland have been published emphasizing the importance of tick-borne pathogens, with relatively low incidence of enteroviruses [8]. However, the clinical burden of meningitis and encephalitis in the pediatric population in Poland remains unknown. Meningitis and encephalitis are important causes of disability-adjusted life years in both children and adults [9]. Neurological sequelae are observed in a substantial number of patients following bacterial meningitis. Viral meningitis was considered a benign illness, but several reports suggest that this might not be the case [1,10]. We, therefore, did an observational study of children admitted with meningitis and encephalitis to determine the incidence, causes and use of antibiotics and antivirals in children with infections of the CNS.

## 2. Patients and Methods

### 2.1. Study Design and Participants

In this retrospective, observational cohort study, children (younger than 18 years) with CNS infection presenting to The Medical University of Bialystok Children’s Clinical Hospital (Bialystok, Poland) between January 2015 and December 2019, were eligible for study inclusion. All the children were hospitalized in the Department of Pediatric Infectious Diseases. Meningitis was defined as symptoms consistent with meningitis and leucocyte count in the cerebrospinal fluid above 5 (children > 1 month) or above 20 (neonates) cells per µL [11]. Encephalitis was defined according to the International Encephalitis Consortium [12] as altered mental status (defined as decreased or altered level of consciousness, including change in personality, lethargy) for over 24 h with no alternative cause identified and 2 of the following: seizures, focal neurologic findings, abnormalities in electroencephalography or magnetic resonance imaging suggestive of encephalitis, cerebrospinal fluid (CSF) pleocytosis and fever. Children who were diagnosed with non-infectious CNS disease, including Guillain-Barré syndrome, acute disseminated encephalomyelitis, Kawasaki disease with meningitis, neoplastic meningitis and drug-induced meningitis were not included in the study as they were hospitalized in different units. Our study did not include neonates hospitalized in neonatal intensive care units.

### 2.2. Microbiological Testing

Cerebrospinal fluid and blood samples were obtained from children if they had clinically suspected meningitis or encephalitis. Blood and CSF samples were examined by aerobic culture in a local laboratory if bacterial CNS infection was suspected. Additionally, those samples were sent to the reference laboratory in Warsaw, Poland to detect *Neisseria meningitidis*, *Haemophilus influenzae*, *Streptococcus pneumoniae*, *Streptococcus pyogenes*, *Streptococcus agalactiae*, *Listeria monocytogenes*, *Escherichia coli* with the use of in-house polymerase chain reaction (PCR). Bacterial CNS infections were diagnosed based on the detection of an appropriate pathogen from either blood or CSF. Children suspected of aseptic CNS infection were routinely tested for enteroviral meningitis, tick-borne encephalitis (TBE) and Lyme neuroborreliosis. Enteroviruses were detected in CSF with diagnostic pan-enterovirus RT-PCR in the majority of cases. From August 2019 on, we used Xpert EV (Cepheid, Sunnyvale, CA, USA). In case of negative CSF test, stool specimens were tested with pan-enterovirus RT-PCR. Tick-borne encephalitis and Lyme neuroborreliosis were confirmed by detection of specific antibodies in serum and CSF using commercially available ELISA kits. Detection of anti-*Borrelia burgdorferi* antibodies was confirmed with Western blot. If the standard testing failed to detect the etiology of aseptic CNS infection, CSF samples were then tested with viral meningitis multiplex PCR (Fast Track Diagnostics, Luxembourg) to detect 11 pathogens: adenovirus, enterovirus, varicella zoster virus, human herpes virus 6, human herpes virus 7, human cytomegalovirus, Epstein-Barr virus, human parechovirus, herpes simplex virus 1, herpes simplex virus 2 and parvovirus B19. Children in whom the above tests were negative were further tested with serological assays to detect antibodies against *Toxocara canis*, *Mycoplasma pneumoniae* and Epstein-Barr Virus. During the influenza season, children were tested for influenza with a rapid antigen test. Non-infectious meningitis or encephalitis was diagnosed if the described work-up did not reveal the cause of the infection, and non-infectious causes were likely when considering patient’s history, laboratory results (mainly an albuminocytological dissociation in the CSF), neurological symptoms, MRI abnormalities and lack of response to antimicrobial treatment. Children without a confirmed etiology of the CNS infection and without signs and symptoms suggestive of non-infectious causes were diagnosed with a CNS infection of unknown cause.

### 2.3. Estimation of Meningitis Incidence

The Medical University of Bialystok Children’s Clinical Hospital hospitalizes all children diagnosed with CNS infections in the North-East Poland. There are several departments for adults in that region, however, children with infections of the CNS are not treated there, for administrative and organizational reasons. We cannot rule out the possibility that a few adolescents were hospitalized there as an exemption. However, throughout the years the number of children with CNS infection hospitalized in our unit was almost equal to the number of officially reported cases in the region. It is therefore possible to estimate incidence rates by dividing the number of patients recruited in the study in 1 year by the total pediatric population of the region. Around 206,000 children younger than 18 years live in the region. Using population data from the Demographic Surveys Department [13] from mid-2017, the estimated incidence was used to calculate population-standardized number of cases in Poland.

### 2.4. Statistical Analysis

Descriptive statistics were used to characterize the cohort of children with CNS infections, with denominators reflecting the number of patients tested or described. Categorical data were compared using Fisher’s exact test for the analysis between two groups or the Kruskal-Wallis test, when comparing multiple groups. Continuous variables were analyzed using Kruskal-Wallis for multiple groups or Mann-Whitney U tests when two groups were compared. To calculate 95% confidence intervals (CIs) for incidence rates we used the Boice-Monson method [14]. Statistical significance was defined as *p* < 0.05. Data were statistically analyzed with TIBCO Software Inc. (2017) Statistica, version 13 (Palo Alto, CA, USA).

### 2.5. Ethical Considerations

The study was conducted in accordance with the Guidelines for Good Clinical Practice. Ethical approval was given by The Bioethical Commission of The Medical University of Bialystok (decision no. APK.002.186.2020).

## 3. Results

### 3.1. Etiology of Meningitis and Encephalitis

Between January 2015 and December 2019, 374 children fitted the definition of CNS infection. 332 (89%) of 374 children were diagnosed with meningitis, and 42 (11%) of them with encephalitis (Figure 1). Clinical features are presented in Table 1.

A total of 305 (82%) of 374 children were diagnosed with an aseptic CNS infection. Viral etiology was confirmed in 278 (74%) of 374 children (Table 2). Enteroviruses accounted for 224 (60%) of all patients. A total of 68 (18%) of 374 cases were tick-borne infections (tick-borne encephalitis virus (*n* = 42) and *Borrelia burgdorferi* (*n* = 26)). Bacterial pathogens were detected in 26 (7%) children, with *Neisseria meningitidis* being the most common bacterial cause, responsible for 9 (35%) of 26 confirmed bacterial cases.

### 3.2. Microbiological Studies

A total of 184 (82%) of 224 cases of enteroviral infection of the CNS were confirmed after detecting the virus in CSF samples. In 40 (18%) of 224 children with enteroviral meningitis CSF PCR was negative for enteroviruses and the diagnosis was made after detecting enteroviruses in stool samples. In 7 (27%) of 26 children with bacterial etiology, CSF and blood cultures were negative, and the diagnosis was made by detection of the pathogen with PCR of CSF or blood samples.

A total of 43 (11%) of 374 children had no cause of CNS infection identified. Of those 12 (28%) had encephalitis and 31 (72%) had meningitis. 12 (28%) had neutrophil predominance in the CSF (>50% neutrophils), and 20 (47%) had a lymphocytic CSF (>50% lymphocytes). In 11 (26%) of 43 children with no identified cause of the infection, the predominant leucocyte type was unknown. We retrospectively re-evaluated laboratory results of those children. We used bacterial meningitis score [15] in children who were not pre-treated with antibiotics. In those receiving antibiotics before CSF collection, we relied on serum inflammatory markers. Considering high serum concentration of C-reactive protein (CRP), elevated procalcitonin, increased white blood count with neutrophil predominance, or absolute number of neutrophils in the CSF above 1000 cells per µL, we classified children as having probably bacterial or probably viral infection of the CNS. A total of 11 (26%) of those 43 children were classified as probably having a bacterial infection, whereas 32 (74%) were classified as probably having a viral infection of the CNS.

### 3.3. Use of Antibiotics and Acyclovir

A total of 16 (6%) of 278 children with confirmed viral CNS infection received a course of antivirals for at least 5 days. All children with VZV infection received acyclovir for a median duration of 16 days (IQR 14–21). Acyclovir was initially administered to four children with TBE and to one with enteroviral CNS infection. All of those five presented symptoms of encephalitis on admittance. The child with HHV-6 encephalitis was treated with ganciclovir for 1 week followed by valganciclovir for 7 weeks. The child with meningitis caused by EBV did not receive antivirals. Nine (21%) of 43 patients with unknown cause of the CNS infection received intravenous acyclovir, all of them were diagnosed with encephalitis.

At least one dose of antibiotics was administered to 23 (8%) of 278 children with proven viral CNS infection. A total of 10 (43%) of them were diagnosed with TBE, 9 (39%) with enteroviral infection, 2 (9%) with VZV infection, 1 (4%) with HHV-6, and 1 (4%) with EBV. Empirical treatment with antibiotics was administered to 24 (56%) of 43 children with an unknown cause of the infection, including 12 (39%) with meningitis and 12 (100%) with encephalitis. A total of 13 (41%) children with CNS infection of unknown, but probably viral cause received antibiotics, including 3 (14%) with meningitis and 10 (100%) with encephalitis. They were 16 times more likely (OR 16.3; 95%CI 6.2–43.2; *p* < 0.001) to receive antibiotics than children with enteroviral infection of the CNS. Excluding cases of encephalitis, children with meningitis of unknown, but probably viral cause were five times more likely (OR 5.6; 95%CI 1.3–24.2; *p* = 0.02) to receive antibiotics than children with enteroviral meningitis.

Children with proven enteroviral infection stayed in hospital shorter than did children with the CNS infections of all other etiologies. The usage of antibiotics was associated with higher median duration of hospital stay in children diagnosed with CNS infection caused by enteroviruses (median 7 (7–9) vs. median 11 (8–13); *p* = 0.003), and in children with probably viral infection of the CNS (median 13 (10–14) vs. median 18 (17–22); *p* < 0.001).

None of the children died before discharge. Seven (2%) of 374 patients required treatment in intensive care.

### 3.4. Incidence and Seasonality of CNS Infections

Using collected data, the incidence of CNS infections in Polish children was calculated (Table 3). Considering all cases, including those with unknown cause, the incidence of meningitis in children younger than 18 was estimated to be 32.22 (95% CI, 25.33–40.98) per 100,000 per year, and that of encephalitis to be 4.08 (95% CI, 2.07–8.02) per 100,000 per year. The highest incidence rate of bacterial CNS infections was observed in children under 1 year of age (22.36 (95% CI, 6.63–75.4) per 100,000 per year), whereas the highest incidence of aseptic meningitis and encephalitis in children between 4 and 6 years (51.82 (95% CI, 32.4–82.9) per 100,000 per year). Using calculated incidence, we estimated the population-standardized number of cases in children in Poland (Table 4).

Hospitalizations for aseptic CNS infections occurred from January to December, however, there was a peak during warm season (Figure 2 and Figure 3). A total of 194 (87%) of 224 cases of enteroviral meningitis and 48 (71%) of 68 CNS infections transmitted by ticks were recorded from June to October. Infections with no identified cause displayed a similar seasonality with 20 (63%) of 32 probably viral cases and 5 (45%) of 11 probably bacterial cases hospitalized from June to October. Five (50%) of 10 probably viral cases of encephalitis were hospitalized from June to August. Bacterial and VZV CNS infections occurred year-round and did not vary by season.

## 4. Discussion

In this study, we have shown that enteroviruses were the main causative agents of the CNS infections, followed by tick-borne infections and VZV. Using clinical data, we estimated the annual incidence of all-cause meningitis in children to be 32.22 (95% CI, 25.33–40.98) per 100,000. In a previous study from the USA, the estimated incidence of meningitis was 27.9 (95% CI, 25.3–30.4) per 100,000 per year, however, the study spans the period before broad introduction of immunization and includes both adults and children [16]. The incidence of encephalitis in our study was estimated to be 4.08 (95% CI, 2.07–8.02) per 100,000 per year, which is lower than previously reported in the USA [17] or Canada [18]. Children with non-infectious causes of encephalitis or meningitis and neonates hospitalized in neonatal intensive care units were excluded from our study, hence, our calculations are probably underestimated. However, a recent Polish study based on laboratory reports reported the annual incidence of meningitis and encephalitis in Polish adult and pediatric population to be between 7.09 and 9.06 per 100,000 (CIs were not calculated in that study), which is significantly lower than our estimates [19]. In that study, enteroviruses constituted only 15.3% of all viral meningitis cases, which probably explains the low reported incidence rates. The clinical course of meningitis caused by enteroviruses is relatively mild and self-limiting [20]. Hence, enteroviral meningitis might be underdiagnosed and underreported in Poland. Broader use of molecular methods in diagnosing etiology of meningitis yields higher numbers of enteroviruses detected. In a prospective study from the UK, where each CSF sample collected was tested with molecular methods, enteroviruses were the most commonly detected viruses in adults [1]. Similarly, in retrospective studies from Greece [21], Sweden [6], UK [22] and Germany [23], enteroviruses were the major causative agents of aseptic meningitis.

The second most common cause of CNS infections were tick-borne encephalitis and Lyme neuroborreliosis, accounting for 22% of all cases of aseptic CNS infections. Poland is considered an endemic area for TBE [24], and Lyme disease [25], but the distribution of the two diseases is focal. The incidence of TBE and Lyme disease in the North-East Poland is the highest among other regions of Poland [8]. In fact, only a limited number of Polish municipalities report TBE cases. Stefanoff et al. argue that, according to environmental investigations and serological surveys, TBE in Poland is underdiagnosed [26,27]. TBEV was responsible for as much as 8% of all meningitis and 31% of all encephalitis cases in our study. Assuming wide-spread distribution of TBE in Poland, our estimates suggest that the number of pediatric cases of TBE should exceed the total number of cases that are currently reported in Poland [8,19]. That implies a concerning conclusion that many pediatric cases remain unrecognized. Although the clinical course of TBE infections is generally less severe in children as compared to adults, the potential for long-term cognitive sequelae in the pediatric population has been recently described [28].

Herpesviruses are an important cause of CNS infections. Herpes simplex virus is frequently detected in adults and in children. The incidence varies according to the geographical region. UK, USA and China reports a high incidence of enteroviruses, whereas Finland has a predominance of herpesvirus meningitis [1,7,22,29,30,31,32]. During the study period we did not identify CNS infections caused by HSV. Still, however, 4% all viral CNS infections were caused by other herpesviruses, mainly by VZV.

In contrast to other studies, a relatively small percentage of our patients had no cause identified. When considering encephalitis separately, we could not determine the cause of 12 (28%) of 42 cases. Despite advances in diagnosing encephalitis, in other studies, up to 80% of cases remained without an identified etiology [17,33,34,35]. Our data suggest that use of molecular methods can increase detection rate of viral and bacterial pathogens in diagnosis of patients hospitalized with infection of the CNS. Management of CNS disease caused by unknown agent poses a challenge to clinicians. It is recommended that patients diagnosed with infectious encephalitis should receive empiric acyclovir and antibacterial agents until bacterial and viral study results are available [36]. Therefore, prompt identification of a specific pathogen reduces the inappropriate use of antibiotics and antivirals and hospital admission costs [1,2,5,20]. In line with these studies, we show that detection of enteroviruses was associated with shorter hospital stay and lower risk of receiving antibiotics. A high percentage of children with an infectious cause identified has probably caused the low use of antibiotics in this study. Still, however, children with unknown cause of the infection were more likely to receive antibiotics, even if they were presented with probably viral meningitis. That indicates a need for an improvement in antibiotic stewardship in our setting.

Interestingly, the majority of cases caused by unknown pathogens were hospitalized in the hot season, characterized by the highest activity of mosquitoes. These results parallel recent studies in the UK and Canada, in which seasonal clusters of encephalitis were reported, suggesting that unknown etiologies of encephalitis may be underlined by unidentified arboviral agents [17,18].

There are limitations to our study. Firstly, studies which rely on retrospective data not originally collected for research purposes are subject to challenges associated with accuracy and completeness of medical records, especially with regard to descriptions of the signs and symptoms of the infection. Secondly, cases of meningitis were presumed to be caused by enterovirus infection if an enterovirus was detected in a stool sample. Enteroviruses are ubiquitous and can be shed with stool without being the cause of disease for prolonged time [37]. However, we did not observe differences in the course of the infection between children diagnosed with the two methods. PCR in stool samples seems to be a reliable alternative to CSF testing in meningitis caused by enteroviruses, which was also shown in other studies [20]. Third, we extrapolated the local incidence of CNS infections in children to calculate annual numbers of cases in Poland, assuming that incidence rates of infections are equal across the country. It is important to note that the number of children living in the region constitutes only 3% of the all pediatric population of Poland. Therefore, our estimates of the total number of pediatric cases in Poland should be interpreted with caution.

## 5. Conclusions

In summary, our study shows that enteroviruses are the major cause of meningitis in children in Poland. Infections transmitted by ticks also pose a significant clinical burden. By the broad use of molecular diagnostic methods, we managed to identify etiology of the CNS infection in the majority of children, reducing inappropriate use of antimicrobials. We suggest that thorough diagnostics of CNS infections are needed to improve management and reduce costs.

## Figures and Tables

**Figure 1 jcm-09-02324-f001:**
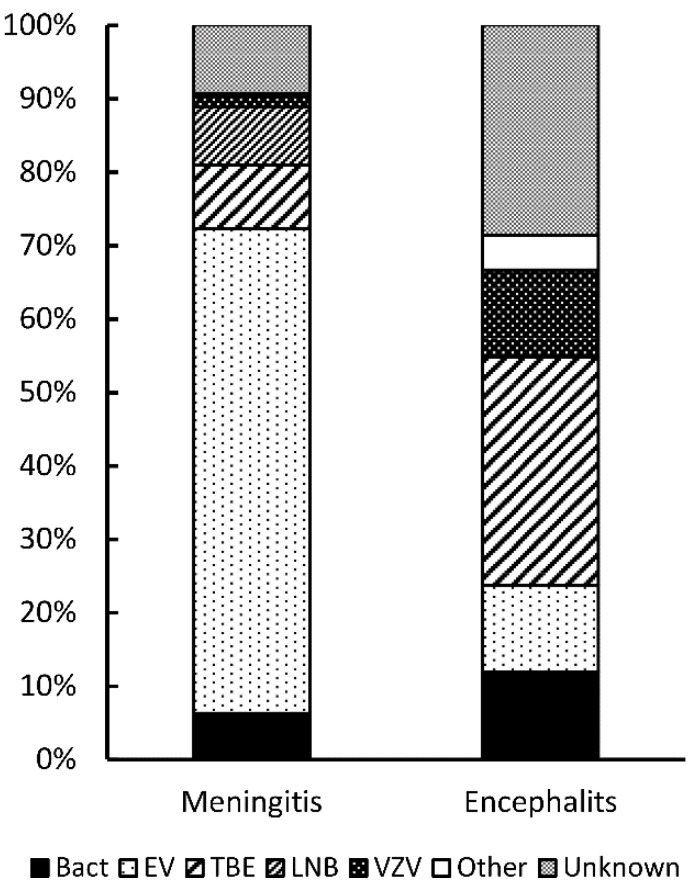
Etiology of infectious meningitis and encephalitis in children included in the study. Abbreviations: Bact: bacterial; EV: enteroviral; TBE: tick-borne encephalitis; LNB: Lyme neuroborreliosis; VZV: varicella zoster virus.

**Figure 2 jcm-09-02324-f002:**
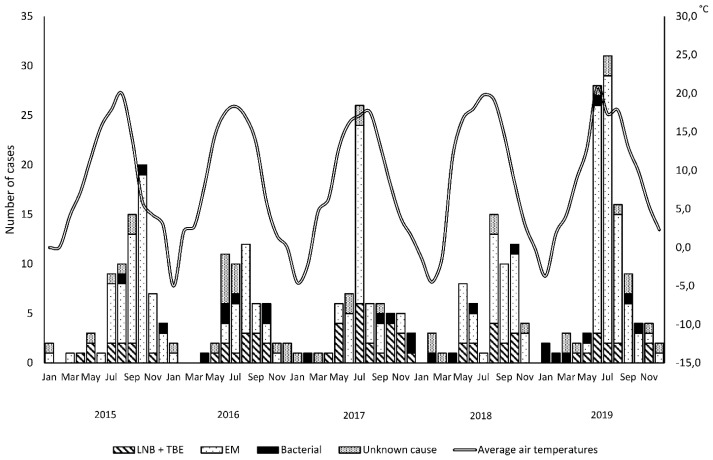
Yearly distribution of cases included in the study. Abbreviations: LNB: Lyme neuroborreliosis; TBE: tick-borne encephalitis; EM: enteroviral meningitis.

**Figure 3 jcm-09-02324-f003:**
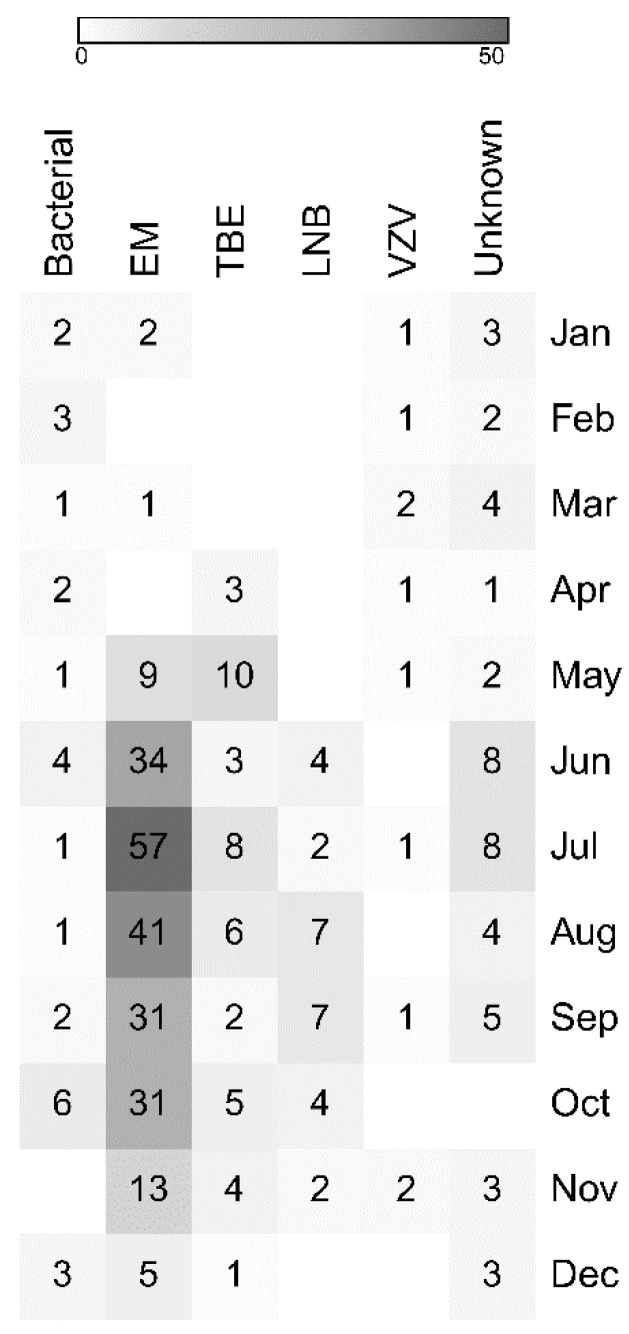
Monthly distribution of cases included in the study shown as a heat map. Abbreviations: LNB: Lyme neuroborreliosis; TBE: tick-borne encephalitis; EM: enteroviral meningitis; VZV: varicella zoster virus.

**Table 1 jcm-09-02324-t001:** Clinical features of study population by cause of the central nervous system (CNS) infection.

	All CNS Infections	All Confirmed Bacterial CNS Infections	All Confirmed Aseptic CNS Infections	All CNS Infections of Unknown Cause		Aseptic CNS Infections				Unknown Cause of CNS Infection		
					*p* *	Enteroviruses	Tick Borne Encephalitis Virus	*Borrelia burgdorferi*	Varicella Zoster Virus	Probably Bacterial CNS Infections	Probably Viral CNS Infections	*p* **
Year												
2015; *n* (%)	76 (100%)	3 (4%)	67 (88%)	6 (8%)	-	54 (71%)	7 (9%)	3 (4%)	3 (4%)	0 (0%)	6 (8%)	-
2016; *n* (%)	54 (100%)	6 (11%)	35 (65%)	13 (24%)	-	22 (41%)	4 (7%)	8 (15%)	0	3 (6%)	10 (18%)	-
2017; *n* (%)	72 (100%)	5 (7%)	60 (83%)	7 (10%)	-	35 (49%)	14 (19%)	7 (10%)	3 (4%)	5 (7%)	2 (3%)	-
2018; *n* (%)	64 (100%)	4 (6%)	54 (84.5%)	6 (9.5%)	-	38 (59%)	8 (13%)	6 (9.5%)	2 (3%)	2 (3%)	4 (6%)	-
2019; *n* (%)	108 (100%)	8 (7.5%)	89 (82.5%)	11 (10%)	-	75 (69%)	9 (8%)	2 (2%)	2 (2%)	1 (1%)	10 (10%)	-
Total; *n* (%)	374 (100%)	26 (7%)	305 (82%)	43 (11%)	-	224 (60%)	42 (11%)	26 (7%)	10 (3%)	11 (3%)	32 (9%)	-
Gender												
Female; *n* (%)	137 (100%)	11 (8%)	107 (78%)	19 (14%)	*NS*	77 (56%)	16 (12%)	9 (7%)	5 (4%)	5 (4%)	14 (10%)	*NS*
Male; *n* (%)	237 (100%)	15 (6%)	198 (84%)	24 (10%)	*NS*	147 (62%)	26 (11%)	17 (7%)	5 (2%)	6 (3%)	18 (8%)	*NS*
Age (years)	8.21 (4.93–13.04)	0.83 (0.15–5.41)	8.8 (5.6–13.7)	6.52 (3–11.39)	<0.001	8.1 (5.4–12.7) ^2^	11.9 (8.5–16.6) ^1^	10.02 (7.42–13.7)	9.1 (4.2–15.6)	4.47 (1.82–5.92)	8.32 (4.29–14.08)	0.02
Age groups												
<1; *n* (%)	22 (100%)	13 (59%)	3 (14%)	6 (27%)	<0.001	3 (14%)	0	0	0	2 (9%)	4 (18%)	*NS*
1–3; *n* (%)	36 (100%)	6 (17%)	25 (69%)	5 (14%)	0.02	18 (50%)	0	3 (8%)	2 (6%)	2 (6%)	3 (14%)	*NS*
4–6; *n* (%)	102 (100%)	2 (2%)	87 (85%)	13 (13%)	0.02	74 (73%)	8 (8%)	3 (3%)	2 (2%)	7 (7%)	6 (6%)	0.009
7–13; *n* (%)	136 (100%)	4 (3%)	121 (89%)	11 (8%)	0.02	89 (65%)	14 (10%)	14 (10%)	3 (2%)	0 (0%)	11 (8%)	0.04
14–17; *n* (%)	78 (100%)	1 (1%)	69 (88%)	8 (10%)	0.02	40 (51%)	20 (26%)	6 (7%)	3 (4%)	0 (0%)	8 (10%)	*NS*
Clinical presentation												
Meningitis; *n* (%)	332 (100%)	21 (6%)	280 (84%)	31 (10%)	*NS*	219 (66%) ^2,3^	29 (9%) ^1,3^	26 (8%) ^1,2^	5 (2%)	9 (3%)	22 (7%)	*NS*
Encephalitis; *n* (%)	42 (100%)	5 (12%)	25 (59%)	12 (29%)	*NS*	5 (12%) ^2,3^	13 (31%) ^1,3^	0^1,2^	5 (12%)	2 (5%)	10 (24%)	*NS*
Signs and symptoms												
Headaches	333/374 (89%)	8/13 (62%)	291/305 (95%)	34/43 (79%)	<0.001	221/224 (98%)	40/42 (95%)	19/26 (73%)	9/10 (90%)	9/11 (82%)	25/32 (78%)	*NS*
Fever	314/374 (84%)	25/26 (96%)	260/305 (85%)	29/43 (67%)	*NS*	203/224 (91%) ^3^	41/42 (98%) ^3^	6/26 (23%) ^1,2^	7/10 (70%)	10/11 (91%)	19/32 (59%)	*NS*
Vomiting	251/373 (67%)	14/26 (54%)	212/305 (69%)	25/42 (59%)	*NS*	170/224 (76%) ^3^	24/42 (57%)	7/26 (27%) ^1,4^	9/10 (90%) ^3^	7/10 (70%)	18/32 (56%)	*NS*
Photophobia	73/373 (20%)	8/26 (31%)	55/305 (18%)	10/42 (24%)	*NS*	42/224 (19%)	7/42 (17%)	2/26 (8%)	3/10 (30%)	3/10 (30%)	7/32 (22%)	*NS*
Neck stiffness	255/374 (68%)	13/26 (50%)	220/305 (72%)	22/43 (515)	0.02	164/224 (73%)	32/42 (76%)	14/26 (54%)	8/10 (80%)	7/11 (64%)	15/32 (47%)	*NS*
Tremor	18/374 (5%)	1/26 (4%)	13/305 (4%)	4/43 (9%)	*NS*	3/224 (1%)	6/42 (14%)	0/26	3/10 (30%)	0/11 (0%)	4/32 (13%)	*NS*
Seizures	15/374 (4%)	3/26 (12%)	5/305 (2%)	7/43 (16%)	0.02	2/224 (1%)	2/42 (5%)	0/26	0/10	0/11 (0%)	7/32 (22%)	*NS*
Facial nerve palsy	23/374 (6%)	0/26 (0%)	21/305 (7%)	2/43 (5%)	*NS*	0/224 ^3^	2/42 (5%) ^3^	19/26 (73%) ^1,2,4^	0/10 ^3^	0/11 (0%)	2/32 (6%)	*NS*
Cognitive dysfunction	31/362 (9%)	8/15 (53%)	16/305 (5%)	7/43 (16%)	<0.001	3/224 (2%)	10/42 (24%)	0/25	1/10 (10%)	2/11 (18%)	5/32 (16%)	*NS*
Altered level of consciousness	66/374 (18%)	20/26 (77%)	31/305 (10%)	15/43% (35%)	<0.001	8/224 (4%) ^2^	20/42 (48%) ^1,3^	0/25^2^	2/10 (20%)	4/11 (18%)	11/32 (34%)	*NS*
Length of hospital stay (days)	10 (7–16)	24 (16–34)	8 (7–13)	15 (13–19)	<0.001	7 (7–9) ^2,3,4^	16 (13–18)^1^	23 (21–25) ^1^	23 (15–24)^1^	17 (14–21)	14 (12.5–18.5)	*NS*
Antibiotics	100/374 (27%)	26/26 (100%)	50/305 (16%)	24/43 (56%)	<0.001	9/224 (4%)	10/42 (24%)	26/26 (100%)	2/10 (20%)	11/11 (100%)	13/32 (41%)	0.003
Acyclovir	26 (7%)	0/26 (0%)	17/305 (6%)	9/43 (21%)	*NS*	1 (<1%)	4 (10%)	0 (0%)	10 (100%)	0/11 (0%)	9/32 (28%)	*NS*
CSF cells (/µL)	144 (48–385)	2347 (288–5800)	136 (49–313)	126 (30–305)	<0.001	135 (46.5–357.5)	87.5 (50–178)	196.5 (62–389)	591 (144–930)	444 (14–1290)	120.5 (42–220)	*NS*
CSF protein (mg/dL)	35 (26–57)	134 (52–283.5)	35 (26–53)	37 (26–72)	<0.001	32.5 (25–43.5) ^2,3^	44 (31–61) ^1^	59.4 (34–83) ^1^	81 (20–170)	36 (21–83)	37 (26–72)	*NS*
CSF lymphocytes (%)	63 (30–83)	14 (8–21)	68 (37–84)	74 (23–90)	<0.001	60 (32–80) ^3,4^	71 (35.5–86) ^3,4^	88.5 (82–94) ^1,2^	91.5 (83–98) ^1,2^	15 (5–55)	78 (34–92)	0.009
CSF neutrophils (%)	23 (6–58)	79 (63–91)	20 (6–52.5)	8.5 (3–73)	<0.001	26 (8–57) ^3,4^	17.5 (7–53.5) ^3^	3 (1–9) ^1,2^	3 (0–11) ^1^	82 (30–89)	6.5 (2–46)	0.008
CSF monocytes (%)	7 (2–12)	5.5 (1.5–15)	7 (2–12)	6 (1–14)	*NS*	7 (2–13)	7 (3–11)	6 (3–11)	3.5 (2–6)	5 (0–8)	6 (2–14)	*NS*
CRP (mg/L)	4.16 (1.2–14.17)	154 (93.32–228.25)	3.36 (1.1–9.46)	7.37 (1.4–58.69)	<0.001	3.79 (1.24–10.88) ^3,4^	5.96 (2.48–12) ^3,4^	0.65 (0.34–1.51) ^1,2^	0.28 (0.15–1.73) ^1,2^	214.42 (59.77–306)	3.21 (0.8–11.46)	<0.001
CRP > 10 mg/L	117/372 (31%)	24/25 (96%)	74/304 (24%)	19/43 (44%)	<0.001	57 (25%)	14 (33%)	2 (8%)	0	11/11 (100%)	8/32 (25%)	<0.001
WBC (×10^9^ cells/L)	9.22 (6.94–12.4)	19.65 (10.94–25.8)	8.8 (6.89–11.34)	12.1 (7.6–17.71)	<0.001	8.8 (6.94–11.07) ^2,3^	11.25 (8.44–15.08) ^1,3,4^	6.59 (5.8–8.8) ^1,2^	7.23 (4.97–8.55) ^2^	23.7 (17.17–35.6)	9.7 (6.89–12.84)	<0.001
Blood Lymphocytes (%)	25 (15–36)	21 (10–37)	25.8 (16.7–36)	24 (12.4–31.4)	*NS*	27.15 (17.65–36.5) ^2^	15 (11–21) ^1,2,4^	32.85 (25.9–41.1) ^2^	32.6 (26.2–51.9) ^2^	11 (6–24)	25.5 (13–37.6)	*NS*
Blood neutrophils (%)	63 (51.5–76.8)	66.55 (56–83)	62 (51.2–74.9)	68.5 (58.1–81)	*NS*	60.9 (50.7–72.8) ^2^	75.8 (66.7–82) ^1,3,4^	50.25 (45–61.3) ^2^	51.5 (30.8–62.6) ^2^	81 (68.5–88)	64.6 (53.9–79)	*NS*

Continuous variables are presented as medians (IQR). Categorical variables are shown as frequencies. *—*p*-value for comparing aseptic and bacterial infections of the CNS; **—*p*-value for comparing probably bacterial and probably viral infections of the CNS; ^1^
*p* < 0.01 vs. enteroviral infection of the CNS; ^2^
*p* < 0.01 vs. tick-borne encephalitis; ^3^
*p* < 0.01 vs. Lyme neuroborreliosis; ^4^
*p* < 0.01 vs. CNS infection cause by varicella zoster virus. The differences between aseptic and bacterial infections of the CNS and between probably bacterial and probably viral infections of the CNS were analyzed with the Fisher’s exact test (categorical variables) or the Mann-Whitney U test (continuous variables). The differences between enteroviral infections of the CNS, tick-borne encephalitis, Lyme neuroborreliosis, and CNS infections caused by varicella zoster virus were analyzed by the Kruskal-Wallis test.

**Table 2 jcm-09-02324-t002:** Etiology of CNS infections in Polish children.

Aseptic CNS Infections	
Enteroviruses	224 (61%)
Tick-borne Encephalitis Virus	42 (11%)
*Borrelia burgdorferi*	26 (7%)
Varicella Zoster Virus	10 (3%)
Epstein-Barr Virus	1 (<1%)
Human Herpes Virus 6	1 (<1%)
*Toxocara canis*	1 (<1%)
**Bacterial CNS infections**	
*Neisseria meningitidis*	9 (2%)
*Streptococcus agalactiae*	6 (2%)
*Escherichia coli*	3 (<1%)
*Streptococcus pneumoniae*	3 (<1%)
Other *Streptococci*	3 (<1%)
*Haemophilus influenzae*	1 (<1%)
*Clostridium difficile*	1 (<1%)
**Unknown cause of the infection**	43 (11%)

**Table 3 jcm-09-02324-t003:** Estimated annual incidence of meningitis and encephalitis per 100,000 population per year based on numbers recruited (95% CI).

	Infections of the CNS—All	Meningitis—All	Encephalitis—All	Confirmed bacterial—All	*Neisseria meningitidis*	Confirmed Aseptic—All	Enteroviral—All	Tick-Borne Encephalitis—All	Lyme Neuroborreliosis—All	Varicella Zoster Virus—All	Unknown Cause—All
<1 year	37.85 (14.87–96.33)	27.52 (9.2–82.32)	10.32 (1.72–61.77)	22.36 (6.63–75.4)	-	5.16 (0.41–64.8)	5.16 (0.41–64.8)	0	0	0	10.32 (1.72–61.77)
1–3 years	22.03 (10.61–45.72)	17.13 (7.48–39.21)	4.89 (1.04–23.05)	3.67 (0.61–21.97)	-	15.3 (6.37–36.74)	11.01 (3.92–30.94)	0	1.84 (0.15–23.05)	1.22 (0.06–27.13)	3.06 (0.43–21.72)
4–6 years	60.76 (39.37–93.76)	55.4 (35.17–87.26)	5.36 (1.24–23.1)	1.19 (0.05–26.42)	-	51.82 (32.4–82.9)	44.08 (26.49–73.36)	4.77 (1.01–22.44)	1.79 (0.14–22.44)	1.19 (0.05–26.42)	7.74 (2.3–26.11)
7–13 years	33.23 (22.82–48.39)	29.57 (19.85–44.04)	3.67 (1.18–11.37)	0.98 (0.11–8.75)	-	29.57 (19.85–44.04)	21.75 (13.67–34.61)	3.42 (1.06–11.04)	3.42 (1.06–11.04)	0.73 (0.06–9.21)	2.69 (0.72–10.08)
14–17 years	33.66 (20.5–55.29)	31.94 (19.19–53.15)	1.73 (0.19–15.45)	0.43 (0.01–34.55)	-	29.78 (17.57–50.47)	17.26 (8.63–34.52)	8.63 (3.24–23)	2.59 (0.43–15.5)	1.29 (0.1–16.26)	3.45 (0.73–16.26)
All age groups	36.30 (28.94–45.53)	32.22 (25.33–40.98)	4.08 (2.07–8.02)	2.52 (1.07–5.96)	0.87 (0.20–3.76)	29.6 (23.03–38.04)	21.74 (16.22–29.13)	4.08 (2.07–8.02)	2.52 (1.07–5.96)	0.97 (0.24–3.88)	4.17 (2.14–8.14)

**Table 4 jcm-09-02324-t004:** Estimated number of cases a year in Poland in children <18.

	Estimated Annual Number of Cases in Children in Poland (95% CI)
All CNS infections	2507 (1999–3145)
All meningitis	2225 (1750–2831)
All encephalitis	282 (143–553)
Total confirmed bacterial CNS infections	174 (73–411)
*Neisseria meningitidis*	60 (14–260)
Total confirmed aseptic CNS infections	2044 (1591–2628)
Enteroviruses	1501 (1120–2012)
Tick-borne encephalitis	281 (143–553)
Lyme neuroborreliosis	174 (73–411)
Varicella Zoster Virus	67 (16–268)
Meningitis and encephalitis of unknown cause	288 (147–562)

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
