# Peer review of "Etiology, Clinical Presentation and Incidence of Infectious Meningitis and Encephalitis in Polish Children"

_jcm, 2020, doi:10.3390/jcm9082324_

Round 1
Reviewer 1 Report
This single centre retrospective observational study of causes of meningitis and encephalitis in children in North East Poland presents a comprehensive and useful analysis which will be of value to those working in Poland and those who encounter patients who have traveled to Poland. Overall the work is well-described, well conducted and limitations have been appropriately recognised
I have only minor comments for the authors to address:
- Given the exclusion of non-infectious causes from this study, the title would be better as "Etiology, clinical presentation and incidence of infectious meningitis and encephalitis in Polish children"
- Abstract - I suggest to remove the text stating that "meningitis and encephalitis are increasingly recognized..." because this is probably not true, and at any rate, unnecessary
- Throughout the manuscript - please give confidence intervals around all incidence estimates
- It is unclear whether preterm neonates in neonatal intensive care units were included in the study. Please clarify this, especially because pre-term infants in NICU are at high risk of meningitis with a variety of organisms. Presumably these infants are not transferred to the referral hospital ward in the same way as older infants and children. If this is the case, please mention this as a caveat for interpretation of the study results.
- Throughout the manuscript it is worth describing the children with enteroviral meningitis based on stool PCR alone as having "presumed enteroviral meningitis". It is well recognised that CSF can be negative whilst enterovirus may be detected in stool, but enteroviruses can also be detected in stool of children without CNS infections, and in healthy controls, and so detection in stool does not prove that it is the cause of the aseptic meningitis
- Table 1 - the way that percentages have been calculated is confusing. Sometimes they have been calculated to sum to 100% down part of a column, sometimes across a row. Since analyses are generally conducted across the rows, I would suggest that percentages should sum across rows. So for example, in 2015 3/76 (4%) CNS infections were bacterial, whereas 67/76 (88%) were aseptic.
- Table 1 - it is unclear why numbers do not sum to the whole in all circumstances - using the example above, only 70/76 subjects are accounted for - so what about the 6 missing?
- Table 1 - it appears that multiple ch-squared tests have been performed for the age group analyses - it would be better to use one contingency table to test the distribution across all of the age groups, resulting in a single p value
- Lines 150-151 - the text states 16 times less likely, but the OR is 16.3 which indicates 16 times more likely. I suspect this is because the calculation has been performed with the enteroviral group as reference. If 16 times less likely, the OR should be around 0.06.
Author Response
The authors would like to thank the reviewer for all the comments and suggestions. The methods, results and discussion sections have been corrected according to the reviewer’s recommendations.
Given the exclusion of non-infectious causes from this study, the title would be better as "Etiology, clinical presentation and incidence of infectious meningitis and encephalitis in Polish children
We agree with the suggestion. The title was modified.
Abstract - I suggest to remove the text stating that "meningitis and encephalitis are increasingly recognized..." because this is probably not true, and at any rate, unnecessary
Corrected.
Throughout the manuscript - please give confidence intervals around all incidence estimates
Confidence intervals were added for every but one incidence rate in Line 198. CIs were not calculated in that study.
It is unclear whether preterm neonates in neonatal intensive care units were included in the study. Please clarify this, especially because pre-term infants in NICU are at high risk of meningitis with a variety of organisms. Presumably these infants are not transferred to the referral hospital ward in the same way as older infants and children. If this is the case, please mention this as a caveat for interpretation of the study results.
That is right, we did not include neonates hospitalized in NICUs. That information now is given in the Methods section and again in the Discussion.
Throughout the manuscript it is worth describing the children with enteroviral meningitis based on stool PCR alone as having "presumed enteroviral meningitis". It is well recognised that CSF can be negative whilst enterovirus may be detected in stool, but enteroviruses can also be detected in stool of children without CNS infections, and in healthy controls, and so detection in stool does not prove that it is the cause of the aseptic meningitis.
We agree that detection of EVs in stool samples is common. In our study children were diagnosed with enteroviral meningitis if presented with signs of meningitis, had CSF pleocytosis, and serology for TBE and LNB was negative. Our observations are that longer time from symptoms onset to CSF collection causes lower EV detection rates in the CSF. Secondly, we frequently hospitalize children with "presumed enteroviral meningitis" (CSF -/ stool+) , who were in close contact with other children with "confirmed enteroviral meningitis" (CSF+). Also, viral typing that we did in another study shows that children with "presumed" and "confirmed" enteroviral meningitis do not differ in terms of EV type. Therefore we assumed that detection of EVs in stool samples was reliable enough to diagnose enteroviral meningitis. However, we are aware of the limitations of that approach, which are mentioned in the manuscript.
Table 1 - the way that percentages have been calculated is confusing. Sometimes they have been calculated to sum to 100% down part of a column, sometimes across a row. Since analyses are generally conducted across the rows, I would suggest that percentages should sum across rows. So for example, in 2015 3/76 (4%) CNS infections were bacterial, whereas 67/76 (88%) were aseptic.
We agree with the suggestion and we modified the table 1. The percentages now are calculated across the rows.
Table 1 - it is unclear why numbers do not sum to the whole in all circumstances - using the example above, only 70/76 subjects are accounted for - so what about the 6 missing?
Actually, the numbers do sum up to the whole. The first row: All CNS infections=76: bacterial=3, aseptic=67, unknown cause=6 etc. We added the word "confirmed" to the column "aseptic" and moved the column "Unknown cause" leftwards for clarity.
Truly there was a mistake in the total number of encephalitis and meningitis cases - the numbers in fact did not sum up to the whole. The error was corrected.
Table 1 - it appears that multiple ch-squared tests have been performed for the age group analyses - it would be better to use one contingency table to test the distribution across all of the age groups, resulting in a single p value
The differences between aseptic and bacterial infections of the CNS and between probably bacterial and probably viral infections of the CNS were analyzed with the Fisher’s Exact Test (categorical variables) or the Mann-Whitney U Test (continuous variables). The differences between enteroviral infections of the CNS, tick-borne encephalitis, Lyme neuroborreliosis, and CNS infections caused by Varicella Zoster Virus were analyzed by the Kruskal-Wallis Test. The description of our statistical methods was improved.
Lines 150-151 - the text states 16 times less likely, but the OR is 16.3 which indicates 16 times more likely. I suspect this is because the calculation has been performed with the enteroviral group as reference. If 16 times less likely, the OR should be around 0.06.
We agree with the suggestion. The sentence was turned around to: "Children with CNS infection of unknown cause were 16 time more likely than children with EV..."
Reviewer 2 Report
In this retrospective single-center cohort study, the authors aimed to analyze etiology, clinical presentation and incidence of meningitis and encephalitis in 374 Polish children. The study design as well as preparation of the manuscript deserve a high rate. No methodological concerns need to be raised. It should be highlighted, that similar studies on the large study groups are lacking. Thus, the authors provide new data in this field.
I have only one minor comment, please explain all abbreviations as they are used for the first time (e.g., HSV, VZV in the Introduction part).
Author Response
The authors would like to thank the reviewer for all the comments and suggestions.
I have only one minor comment, please explain all abbreviations as they are used for the first time (e.g., HSV, VZV in the Introduction part).
All the abbreviations were explained on the first mention.
Reviewer 3 Report
Dear Authors,
Overall this is an interesting and reasonably well written article, which I enjoyed reading. It could benefit from some improvements in grammar and clarifications in methods and results, as detailed below.
Kind Regards,
- Line 52 Methods: Any information on treatment guidelines? Was there an antibiotic or institutional guideline in place and was it followed? Similarly there seems to be a lack of information on use of corticosteroids? Was there any use?
- Lines 57: please provide a reference for these definitions/thresholds
- At lines 62-66: it is stated those with non-infectious diagnosis were excluded as they were hospitalized in different units. Was this study focused only on admissions under a single unit? i.e. infectious diseases? This could be clarified.
E.g. what about patients with an uncertain diagnosis? Which unit were they hospitalised under. This may impact on the interpretation of results at ____
- At lines 67-82: Regarding the testing algorithm, what proportion of patients had testing performed exactly according to the algorithm? If not 100% it may be useful to provide more details e.g. a testing ‘cascade’=
- At lines 69-71: Regarding PCR for multiple potential bacterial pathogens. Please clarify if this was on CSF ‘samples’ only? Also, was this performed on a specific CSF multiplex platform? If so, the name of the platform should be mentioned. Also please specific culture method used.
- At lines 77-78: in regards to “specific antibodies” please provide more detail as to the method and/or diagnostic criteria applied.
- At lines 84-85: please clarify, is the Childrens hospital the only facility servicing the area? i.e. is it possible that children could be hospitalised elsewhere? This could be clarified.
- At lines 87: please provide the citation for the number of children in the region?
- Figure 1 caption: please specify this is etiology of (suspected and confirmed?) infectious meningitis and encephalitis only (as it was stated that non-infectious cases were excluded).
- Table 1 and lines 132-13: Please define “probably bacterial” and “probably viral” infections if any specific criteria or scoring system was used? If it was an overall clinician impression/opinion, please state this.
- Table 2: the % figure for Neisseria Meningitidis is messing
- At lines 128: For those with no cause found, at what stage did they go on to have confirmatory testing for non-infectious cases? How does this fit into those who were excluded from the outset (lines 62-66)?
- At lines 231-234: It may be interesting to provide more comments on antibiotic use in the study – was it felt to be overall appropriate? Are there any stewardship observations or improvements to be made.
- At lines 235-238: were the seasonal unknown pathogen cases mostly encephalitis or meningitis cases? The hypothesis is entirely speculative.
Author Response
The authors would like to thank the reviewer for all the comments and suggestions. The methods, results and discussion sections have been corrected according to the reviewer’s recommendations.
Line 52 Methods: Any information on treatment guidelines? Was there an antibiotic or institutional guideline in place and was it followed? Similarly there seems to be a lack of information on use of corticosteroids? Was there any use?
The detailed analysis of treatment protocols was not in the scope of this study. We did not collect data on which antibiotic was used, nor if glucocorticoids were administered. Generally, in our department we follow national recommendations on antibiotic use in bacterial CNS infections. We use ampicillin + cefotaxime and/or gentamicin and/or vancomycin in neonates. For older children we use cefotaxime with vancomycin. Majority of children with bacterial CNS infections receive glucocorticoids. Also, GCs are used in children with encephalitis.
Lines 57: please provide a reference for these definitions/thresholds
References are provided
At lines 62-66: it is stated those with non-infectious diagnosis were excluded as they were hospitalized in different units. Was this study focused only on admissions under a single unit? i.e. infectious diseases? This could be clarified.
E.g. what about patients with an uncertain diagnosis? Which unit were they hospitalised under. This may impact on the interpretation of results at ____
All the children were hospitalized in the Department of Pediatric Infectious Diseases in the University Hospital. That information is now provided in the text. All the children suspected of infection of the CNS are hospitalized in that department first. If the work-up points to non-infectious causes, children who require prolonged hospitalization are transferred to different units within the same hospital. Often children with neurological diseases (like Guillain-Barre syndrome, epilepsy), metabolic disorders or malignancies who are not suspected of CNS infection (based on CSF results, history, clinical presentation) are not hospitalized in the PID department at all. They are treated in the same hospital but in different units, or in the neighboring hospital. We did not have an access to their records, so we assumed that excluding non-infectious meningitis would be better than taking the risk of underestimating their numbers.
At lines 67-82: Regarding the testing algorithm, what proportion of patients had testing performed exactly according to the algorithm? If not 100% it may be useful to provide more details e.g. a testing ‘cascade’=
Our approach is that we actively and thoroughly investigate cases of CNS infections. In every case suspected of bacterial infection we use cultures and PCRs. Similarly, in each case of aseptic meningitis we aim to use all the described methods. However, it might have happened that we lacked CSF for testing in some (mainly infants), hence CSF testing was not done exactly to the described algorithm (but blood samples were, as they are more available). We agree that providing a testing cascade would be an asset to this work, however, that would require reviewing patient's records again, which we cannot afford, unfortunately.
At lines 69-71: Regarding PCR for multiple potential bacterial pathogens. Please clarify if this was on CSF ‘samples’ only? Also, was this performed on a specific CSF multiplex platform? If so, the name of the platform should be mentioned. Also please specific culture method used.
Bacterial pathogens were looked for in both the CSF and the blood. The sentence was rephrased for clarity. The reference lab uses in-house PCR methods. They run PCR reactions with a single or two primers at a time. The culture was aerobic, samples were collected into liquid media.
At lines 77-78: in regards to “specific antibodies” please provide more detail as to the method and/or diagnostic criteria applied.
We used ELISA and Western blotting. The study spans the period of 5 years. During that time our labs changed their diagnostic kits several times. It is impossible to follow exactly when and what changes were made. However, each time the methods were considered reliable.
At lines 84-85: please clarify, is the Childrens hospital the only facility servicing the area? i.e. is it possible that children could be hospitalised elsewhere? This could be clarified.
The PID department hospitalizes all children diagnosed with CNS infections in the North-East Poland. There are several ID departments for adults in that region, but they are very reluctant to admit children for administrative and organizational reasons. We cannot rule out the possibility that they hospitalized a few adolescents. However, when comparing our figures with the number of reported cases we can see that we are the only PID unit reporting pediatric CNS infections in the region. The information was clarified in the text.
At lines 87: please provide the citation for the number of children in the region?
The citation was provided.
Figure 1 caption: please specify this is etiology of (suspected and confirmed?) infectious meningitis and encephalitis only (as it was stated that non-infectious cases were excluded).
The word "infectious" was added.
Table 1 and lines 132-13: Please define “probably bacterial” and “probably viral” infections if any specific criteria or scoring system was used? If it was an overall clinician impression/opinion, please state this.
We used bacterial meningitis score in children who were not pre-treated with antibiotics. In those receiving antibiotics before CSF collection, we relied on serum inflammatory markers and our overall impression. The text was rewritten for clarity.
Table 2: the % figure for Neisseria Meningitidis is messing
Thank you for noticing. The mistake was corrected.
At lines 128: For those with no cause found, at what stage did they go on to have confirmatory testing for non-infectious cases? How does this fit into those who were excluded from the outset (lines 62-66)?
This is a retrospective study. It is difficult to describe stages of the work-up in a retrospective study, as no standard approach was used. In our opinion non-infectious causes have a bit distinct clinical picture. Children were suspected of non-infectious causes if patient's history, neurological symptoms, and MRI abnormalities raised that suspicion, which was also strengthened by the high CSF protein, albuminocytological dissociation, high CSF IgG, and lack of response to antimicrobial treatment.
The paragraph on microbiological testing was completed for better clarity.
At lines 231-234: It may be interesting to provide more comments on antibiotic use in the study – was it felt to be overall appropriate? Are there any stewardship observations or improvements to be made.
As mentioned before, we unfortunately did not look into antibiotic therapy in detail. My personal impression is that use of antibiotics in children with CNS infections followed the guidelines in majority of cases. The important observation is that children with probably viral infection (based on our overall impression) received antibiotics, what extended their hospital stay. That is the place for improvement.
At lines 235-238: were the seasonal unknown pathogen cases mostly encephalitis or meningitis cases? The hypothesis is entirely speculative.
Yes, that is just a hypothesis, but similar observations were made in Canada and the UK. In the hot season we hospitalized 50% cases of encephalitis and 63% cases of meningitis of unknown, but probably viral cause. That information was provided in lines 202-206.